# Association of Vitamin D Deficiency with Local Muscle–Fat Ratio in Geriatric Palliative Care Patients: An Ultrasonographic Study

**DOI:** 10.3390/healthcare13172188

**Published:** 2025-09-01

**Authors:** Ayfer Durak, Umut Safer

**Affiliations:** 1Division of Geriatrics, Department of Internal Medicine, Sancaktepe Prof. Dr. İlhan Varank Training and Research Hospital, Sancaktepe, 34785 Istanbul, Türkiye; umutsafer@gmail.com; 2Division of Geriatrics, Department of Internal Medicine, Amasya University Sabuncuoğlu Şerefeddin Training and Research Hospital, 05100 Amasya, Türkiye

**Keywords:** muscle–fat ratio, palliative care unit, ultrasonography, vitamin D

## Abstract

**Background/Objectives:** Vitamin D deficiency is linked to muscle loss and fat changes in older adults, but data regarding palliative patients are limited. Ultrasound offers a practical tool to assess these changes. This study explores the relationship between vitamin D levels and ultrasound-measured muscle, fat, and their ratio in older adult palliative patients. **Methods:** This prospective cross-sectional study was conducted in a tertiary palliative care unit (June–September 2024). A total of 187 patients were grouped by serum vitamin D levels (<50 vs. ≥50 nmol/L). Demographic and clinical variables included sex, BMI, Activities of Daily Living (ADLs), calf circumference (CC), and comorbidities. Ultrasonography assessed muscle thickness (MT), subcutaneous fat thickness (SFT), and cross-sectional area (CSA) of Rectus Femoris (RF) and Biceps Brachii (BB). MT/SFT ratio was calculated. Logistic regression identified independent predictors. **Results:** Mean age was 75.1 ± 14.4 years; 55.6% of participants were female. Vitamin D deficiency (67.9%) was significantly associated with female sex (*p* = 0.037), ADL dependency (*p* < 0.001), lower BMI (*p* = 0.020), and reduced CC (*p* = 0.006). RF-MT, RF-SFT, RF-CSA, BB-MT, and BB-CSA were lower in the deficient group. RF-MT/SFT ratio was higher (*p* = 0.049). ADL dependency (*p* = 0.002) and RF-MT/SFT (*p* = 0.015) were independent predictors. **Conclusions:** Vitamin D deficiency was linked to a higher muscle-to-fat ratio, mainly due to fat loss rather than muscle gain. This may misrepresent muscle preservation and should be interpreted cautiously. Although vitamin D levels appear to be associated with physical function, additional prospective cohort and interventional supplementation studies are warranted to determine whether routine screening and targeted vitamin D supplementation can effectively support physical function in this population.

## 1. Introduction

Body composition refers to the relative distribution of muscle, fat, and bone tissues in the human body. The proportion of these components is closely associated with an individual’s overall health status, functional capacity, and nutritional habits [1]. In hospitalized individuals, these proportions may shift significantly due to disease progression, physical inactivity, and malnutrition [2]. Furthermore, among the indicators of body composition, the muscle-to-fat ratio has been shown to be significantly associated with adverse clinical outcomes, including functional impairment, increased risk of falls, and the development of complications [2]. Therefore, current sarcopenia guidelines, such as those of the European Working Group on Sarcopenia in Older People (EWGSOP), recommend regular assessment of muscle mass and body composition in hospitalized individuals [3]. In palliative care settings, patients often face advanced-stage chronic illnesses. In this context, preservation of muscle and fat reserves plays a critical role in maintaining physical functionality, preventing pressure ulcers, and improving overall quality of life [4].

In this assessment process, ultrasonography stands out as a practical, portable, non-ionizing, and cost-effective method. It offers high accuracy in measuring muscle thickness and subcutaneous fat thickness, particularly in muscle groups such as the quadriceps femoris and biceps brachii [5]. The muscle-to-fat ratio obtained by evaluating these parameters together is considered an important reflection of body composition. Fried et al. [6] proposed that the fat-to-muscle ratio calculated using ultrasonography can provide meaningful information regarding an individual’s nutritional status and muscle reserves. Although a specific cutoff value for this ratio has not yet been established, this approach has been shown to support more personalized treatment planning [7].

On the other hand, vitamin D, a modifiable biomarker, plays a dual biological role in the regulation of body composition. The presence of vitamin D receptors (VDRs) in skeletal muscle cells indicates that this vitamin is particularly effective in the preservation of type II muscle fibers and the stimulation of protein synthesis [8]. This effect has also been supported by systematic reviews demonstrating the direct effects of vitamin D on skeletal muscle regeneration, hypertrophy, and satellite cell activity in both animal and human models [9]. Additionally, vitamin D is involved in fat metabolism and interacts with adipose tissue. This interaction suggests that vitamin D can influence both muscle and fat tissues, thereby acting as a regulatory factor in body composition [10]. Granic et al. (2017) [11] conducted a prospective cohort study (Newcastle 85+ Study), which demonstrated a significant association between low serum 25(OH)D concentrations and accelerated decline in muscle strength in the very old patients, particularly in men. These findings underscore the importance of maintaining adequate vitamin D status for musculoskeletal health and highlight a critical gap in the current literature regarding its role in sarcopenia and functional decline among the oldest old [11].

Moreover, vitamin D is a fundamental nutrient that affects not only muscle strength but also bone health, immune function, cardiovascular health, cancer, diabetes, and cognitive function [12].

Vitamin D deficiency has been associated with muscle loss and altered fat distribution in older adults [13]. However, evidence on this association in geriatric patients receiving palliative care remains limited, particularly when body composition is assessed using ultrasound-based parameters such as muscle thickness, subcutaneous fat thickness, and muscle-to-fat ratio. In the present study, the term muscle-to-fat ratio (MFR) specifically refers to the ratio of ultrasound-measured muscle thickness to subcutaneous fat thickness at defined anatomical sites (e.g., quadriceps, biceps), rather than the conventional whole-body muscle mass to fat mass ratio. Vitamin D has been extensively studied as a modifiable and easily accessible biomarker. However, in certain high-risk subgroups such as geriatric palliative care patients, its assessment may at times not be prioritized due to the presence of other urgent clinical concerns.

Therefore, this study aims to evaluate the association between serum vitamin D levels and ultrasound-assessed body composition in older adult patients under palliative care.

## 2. Materials and Methods

### 2.1. Study Design and Setting

This prospective observational study was conducted in a tertiary palliative care unit between June and September 2024. The study adhered to the principles of the Declaration of Helsinki and received approval from the local ethics committee (Approval No: 2024/168). Written informed consent was obtained from all participants or their legal representatives. No compensation or financial incentives were offered.

### 2.2. Participants and Eligibility Criteria

Participants were patients admitted to the palliative care unit. All anthropometric and ultrasonographic measurements were performed on the right side of the body to ensure standardization. Exclusion criteria included conditions that could interfere with these assessments, such as right-sided hemiplegia, limb amputation, contractures, or clinically edema (as fluid accumulation can affect body measurements), fractures, burns, or skin disorders. Based on serum 25-hydroxyvitamin [25(OH)D] levels, participants were categorized into two groups according to the Endocrine Society criteria: deficient (<50 nmol/L) and sufficient (≥50 nmol/L) [14] (Figure 1).

### 2.3. Sample Size Calculation

The required sample size was calculated using G*Power software (version 3.1), based on an effect size of Cohen’s d = 0.37, which was derived from a previously published study [10]. In that study, a statistically significant difference was observed in anterior forearm subcutaneous fat thickness between individuals with and without vitamin D deficiency (*p* = 0.021). An independent-sample *t*-test was identified as the appropriate statistical test for this comparison. To achieve a statistical power of 0.80 with a two-tailed alpha level of 0.05, the minimum required sample size was estimated to be 182 participants. To account for a potential data loss of up to 10%, a total of 200 participants were initially recruited. After applying the predefined exclusion criteria, 187 participants were included in the final analyses.

### 2.4. Variables and Measurements

#### 2.4.1. Demographic and Clinical Status

Age, sex, functional capacity, and major comorbidities including common conditions such as diabetes, cancer, stroke, dementia, and hypertension were recorded. Functional capacity was evaluated using the Katz Index of Independence in Activities of Daily Living (ADLs), encompassing bathing, dressing, toileting, transferring, continence, and feeding. Each activity was scored dichotomously (0 = dependent, 1 = independent), with total scores ranging from 0 to 6 [15]. Participants with a total Katz score of 6 were categorized as independent, while those scoring less than 6 were considered dependent.

#### 2.4.2. Anthropometric Measurement

All anthropometric measurements were conducted using calibrated, high-precision instruments (±0.1 kg for weight; ±1 cm for circumference), and all devices were calibrated at regular intervals.

Bedridden patients were weighed using a sling scale (Seca, Model 985, Seca GmbH, Hamburg, Germany), and height was measured in a lateral or supine position using a non-elastic tape (Seca 201, Seca GmbH, Hamburg, Germany). For ambulatory patients, weight was measured using a digital weighing scale (TANITA BC-418, Tanita Corp., Tokyo, Japan), and standing height was measured without shoes using a portable stadiometer (Seca 213, Seca GmbH, Hamburg, Germany).

Body mass index (BMI) was calculated as weight in kilograms divided by height in meters squared (kg/m^2^), with age-specific cutoff values based on the ESPEN guidelines. A BMI of <20 kg/m^2^ was considered low for individuals under 70 years of age and <22 kg/m^2^ was considered low for those aged 70 years or older [16].

Mid-upper arm circumference (MUAC) and calf circumference (CC) were measured on the right side using a non-elastic tape (Seca 201, Seca GmbH, Hamburg, Germany). MUAC was recorded at the midpoint between the acromion and olecranon [17]; CC was measured at the widest point of the calf with the knee flexed at 90° [18]. The average of three consecutive measurements was used for analysis.

Based on the cutoff values proposed by the European Working Group on Sarcopenia in Older People (EWGSOP2), low muscle mass was defined as mid-upper arm circumference (MUAC) of ≤28.6 cm for men and ≤27.5 cm for women, and calf circumference (CC) of <31 cm [3].

#### 2.4.3. Ultrasonographic Muscle–Fat Ratio

Ultrasound-based measurements of muscle and fat were performed in B-mode using a linear transducer (12–4 MHz, Philips Affiniti 50, Philips Healthcare, Andover, MA, USA), following the recommendations of the Sarcopenia Special Interest Group of the European Union Geriatric Medicine Society [19].

All assessments were carried out by a single experienced clinician blinded to the participants’ clinical information. To minimize tissue compression, an appropriate amount of gel was applied, and the probe was positioned perpendicular to the muscle fibers. Subcutaneous fat thickness (SFT), muscle thickness (MT), and cross-sectional area (CSA) of the rectus femoris (RF) and biceps brachii (BB) muscles were measured on the right side of the body with patients in the supine position and limbs fully extended after a 5 min rest. RF was assessed at the distal one-third between the anterior inferior iliac spine and the superior border of the patella [20], while BB was measured at the midpoint between the acromion and the olecranon along the humerus [21] (Figure 2).

SFT was defined as the distance between the skin and the superficial fascia (cm), MT was defined as the distance between the superficial and deep fascia (cm), and CSA was defined as the area enclosed by these two fascia layers (cm^2^). Each parameter was measured three times, and the mean value was used for analysis. The muscle-to-fat ratio was calculated by dividing MT (cm) by SFT (cm) at the same anatomical site.

Intraobserver reliability was evaluated using intraclass correlation coefficients (ICCs) derived from repeated measurements taken 15 min apart in a healthy subsample (n = 15). The resulting ICC values demonstrated excellent reproducibility: RF-MT = 0.95, RF-CSA = 0.93, BB-MT = 0.90, and BB-CSA = 0.95.

#### 2.4.4. Vitamin D Measurement

Serum 25(OH)D levels (nmol/L) were measured from venous blood samples collected in the morning after at least 8 h of fasting, using a colorimetric assay on a Cobas 6000 autoanalyzer (Roche Diagnostics, Mannheim, Germany).

### 2.5. Statistical Analysis

The normality of continuous variables was assessed using the Kolmogorov–Smirnov test. Normally distributed data are presented as mean ± standard deviation, whereas non-normally distributed data are shown as medians (min–max). Categorical variables are expressed as n(%). Group comparisons were made using Student’s *t*-test for normally distributed data, the Mann–Whitney U test for non-normally distributed data, and the Chi-square test for categorical data. Multivariable logistic regression analysis was performed using the Forward Likelihood Ratio method to identify independent predictors of vitamin D deficiency. To assess potential multicollinearity, correlation analyses were conducted. Strong correlations were observed between RF-MT and RF-CSA (r = 0.836, *p* < 0.001) and between BB-MT and BB-CSA (r = 0.824, *p* < 0.001). Therefore, to avoid redundancy and ensure model stability, only age, sex, ADL dependency, hypertension, RF-MT/SFT ratio, and BB-MT were retained as representative independent variables in the final model. Probability estimation curves were generated for BB-MT, which was significant in the univariable analysis and represents a continuous, clinically interpretable measure of muscle thickness.

The results are presented as odds ratios (ORs) with 95% confidence intervals (CIs). Model fit was verified by the Hosmer–Lemeshow test (*p* = 0.931). Analyses were conducted using IBM SPSS Statistics v27.3.

## 3. Results

### 3.1. Participant Characteristics

Of the 220 patients initially screened, 187 met the inclusion criteria and were included in the final analysis. The mean age was 75.1 ± 14.4 years, and 55.6% of participants were female. Based on serum 25(OH)D concentrations, 67.9% of participants were classified as vitamin D deficient (<50 nmol/L), while 32.1% were in the sufficient group (≥50 nmol/L) (Table 1).

### 3.2. Group Comparisons

Vitamin D deficiency was significantly more prevalent among female participants (*p* = 0.037) and those with greater functional dependence, as indicated by lower Katz ADL scores (*p* < 0.001). Deficient individuals also had significantly lower body mass index (BMI) (*p* = 0.020) and calf circumference (*p* = 0.006).

Ultrasonographic assessments showed significantly reduced values for rectus femoris subcutaneous fat thickness (RF-SFT), muscle thickness (RF-MT), cross-sectional area (RF-CSA), and biceps brachii muscle parameters (BB-MT and BB-CSA) in the deficiency group. Most ultrasonography parameters differed significantly between groups (*p* < 0.05), but BB-SFT (*p* = 0.210) and BB-MT/BB-SFT (*p* = 0.968) did not. Notably, the RF-MT/SFT ratio was significantly higher in the deficiency group (*p* = 0.049). Hypertension prevalence demonstrated borderline significance (*p* = 0.050) (Table 2).

### 3.3. Multivariable Analysis

In multivariable logistic regression (adjusted for age, sex, ADL dependency, hypertension, RF-MT/SFT ratio, and BB-MT), two independent predictors of vitamin D deficiency were identified: ADL dependency (OR: 8.86; 95% CI: 2.28–34.43; *p* = 0.002) and RF-MT/SFT ratio (OR: 2.70; 95% CI: 1.21–6.03; *p* = 0.015) (Table 3).

### 3.4. Probability Estimation

An inverse association was found between BB-MT and the probability of vitamin D deficiency. Predictive modeling showed that the likelihood of deficiency exceeded 80% in individuals with low BB-MT, decreasing to approximately 35% in those with higher values (Figure 3).

## 4. Discussion

Vitamin D deficiency is a significant factor that negatively affects musculoskeletal health, particularly in older adults. In this study, we found that the muscle-to-fat ratio was higher in patients with low vitamin D levels. However, this was likely due to a relatively greater reduction in subcutaneous fat compared to muscle thickness, rather than a true improvement in body composition. In the context of palliative care, where both muscle and fat loss are common due to catabolic states and nutritional deficits, this finding highlights the need for cautious interpretation of body composition ratios. An increased ratio in this setting may mask underlying tissue depletion and should not be misinterpreted as preserved muscle mass or improved functional status.

Advanced age, multiple comorbidities, prolonged immobility, and inadequate nutrition can contribute to both muscle loss and impaired vitamin D metabolism [22]. These risk factors are commonly observed in patients receiving palliative care. In our study, the prevalence of vitamin D deficiency, defined as serum 25(OH)D levels below 50 nmol/L according to the Endocrine Society’s threshold, was found to be 67.9%. This rate was lower than those reported in previous studies involving frail individuals (87.4%) [23] and hospitalized older adults (98.7%) [24]. For comparison, a large-scale Romanian study by Bucurica et al. (2023) reported a deficiency rate of 28.8% among 11,182 hospitalized patients, using the same threshold of <50 nmol/L (20 ng/mL) [25]. This difference may be explained by the greater heterogeneity of our patient population or the possibility of prior vitamin D supplementation. This highlights the importance of individualized evaluation of vitamin D status in palliative care settings.

Patients with vitamin D deficiency were predominantly female, demonstrated higher levels of functional dependency (ADL), and had lower BMI and calf circumference values. It is thought that this sex difference may be due to factors such as greater vitamin D storage in adipose tissue in women [26] and hormonal changes after menopause [27]. Physical inactivity may negatively affect vitamin D synthesis by reducing sun exposure. Moreover, a bidirectional relationship is thought to exist between vitamin D and physical performance: adequate vitamin D levels can enhance physical performance, while good physical performance may help maintain sufficient vitamin D levels [28]. Vitamin D deficiency particularly affects type II fast-twitch muscle fibers, which may lead to changes in muscle structure such as stiffness, increased opacity, and fibrosis [8]. Although some studies have shown that supplementation can enhance the expression of vitamin D receptors in muscles and positively influence muscle fiber size, significant effects on physical function [29] have not always been consistently reported [30]. In our study, the higher level of functional dependency observed in individuals with vitamin D deficiency supports the protective effect of this vitamin on the musculoskeletal system.

While vitamin D deficiency in the general population is typically associated with increased adipose tissue [26,31], the situation differs in palliative care patients. Conditions such as malnutrition, unintentional weight loss, dysphagia, pressure ulcers, and systemic inflammation may lead to a loss of adipose tissue [32]. Therefore, the relationship between vitamin D and adipose tissue in this patient group may differ from that observed in healthy individuals. Hypertension, on the other hand, has been significantly associated with vitamin D deficiency in previous studies [33]. However, in our study, this association was only borderline significant and was more evident among individuals with normal vitamin D levels. This finding may be related to confounding factors such as comorbidities, treatment differences, or vitamin D supplementation.

Calf circumference measurement, an indirect indicator of muscle mass [3], along with ultrasound assessments of muscle and subcutaneous fat thickness, provide valuable information about nutritional and functional status. Subcutaneous adipose tissue is not only an energy reservoir but also a critical tissue that helps maintain body temperature, provides cushioning against external forces, and contributes to hormonal regulation [34]. Loss of this tissue may increase the risk of systemic complications, particularly in palliative care patients [35].

In our study, both muscle and subcutaneous fat thickness were found to be significantly reduced in individuals with vitamin D deficiency. Additionally, an increased ratio of rectus femoris muscle thickness to subcutaneous fat thickness (RF-MT/SFT) and the level of functional dependency were independently associated with vitamin D deficiency.

Although some previous studies have associated vitamin D deficiency not only with muscle loss [10] but also with fat accumulation [10], our findings indicate a more pronounced reduction in adipose tissue. This suggests that in palliative care patients with vitamin D deficiency, both muscle and fat tissues may be lost simultaneously, reflecting an ongoing catabolic process. In this context, factors such as inflammation, reduced mobility, and inadequate nutrition appear to play a central role [32]. It is important to emphasize that a high muscle-to-fat ratio does not always indicate better muscle mass; in some cases, it may simply reflect a greater loss of fat tissue. Therefore, this ratio should be interpreted with caution and should not be assumed to directly correspond to functional capacity.

Vitamin D influences various mechanisms in both muscle and adipose tissues, including cell proliferation, differentiation, and inflammation control. Low levels of vitamin D may impair these processes, potentially accelerating the loss of both muscle and fat [8]. These effects can become even more pronounced in individuals with chronic illnesses or those under catabolic stress [36].

Our study also has several limitations. Its single-center and cross-sectional design prevents us from establishing causal relationships. The lack of sex-stratified analyses is acknowledged as a limitation of this study. Palliative care population is highly heterogeneous, and tissue loss may be influenced by numerous variables. Potential confounding factors such as vitamin D supplementation, medications, and renal function could not be controlled. Furthermore, nutritional status could not be systematically assessed using validated clinical tools or dietary records in this study. Accordingly, variations in caloric intake, protein consumption, and overall nutritional support might have influenced body composition and muscle loss. Additionally, although functional status was assessed using the KATZ index and included in the analysis as a confounding variable (dependent/independent status), we did not perform a detailed analysis of activity levels across all participants. Therefore, the potential impact of varying degrees of physical activity on the outcomes could not be fully evaluated. Logistic regression analysis can be considered an appropriate choice given the categorical nature of vitamin D status. However, this method may be insufficient to fully reflect potential physiological causal relationships between variables. In addition, the forward likelihood ratio (LR) method used for variable selection has certain statistical limitations [37]. Therefore, future studies would benefit from employing more robust model selection criteria, such as the Akaike Information Criterion (AIC), and from planning longitudinal designs that use linear modeling approaches to better assess causal relationships.

Nevertheless, we believe that our study makes a significant contribution as one of the first to examine the relationship between vitamin D deficiency and body composition in palliative care patients using ultrasound-based muscle-to-fat ratio assessment. In this regard, it may offer a novel perspective to the existing literature.

## 5. Conclusions

Our findings suggest that vitamin D deficiency may be associated with alterations in body composition among palliative care patients, particularly a higher muscle-to-fat ratio primarily driven by fat loss rather than muscle preservation. This finding should therefore be interpreted with caution, as it may misrepresent muscle status. Moreover, given the cross-sectional design of our study, a causal relationship cannot be established. To clarify the potential effects of vitamin D supplementation on physical function, further prospective cohort studies and interventional trials are warranted.

## Figures and Tables

**Figure 1 healthcare-13-02188-f001:**
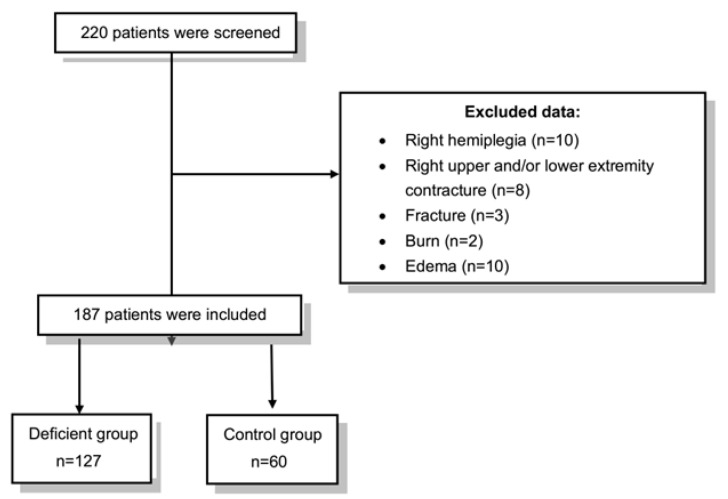
A flow diagram of the patient selection procedure.

**Figure 2 healthcare-13-02188-f002:**
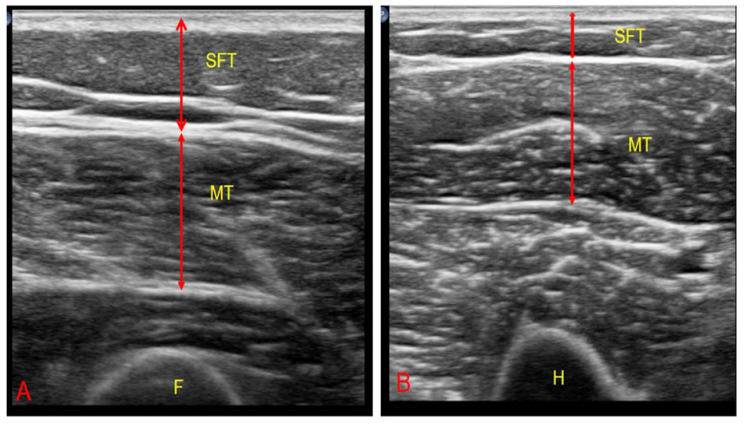
(**A**) Transverse ultrasonographic image of the rectus femoris muscle. (**B**) Transverse ultrasonographic image of the biceps brachii muscle. F, femur; H, humerus; MT, muscle thickness; SFT, subcutaneous tissue.

**Figure 3 healthcare-13-02188-f003:**
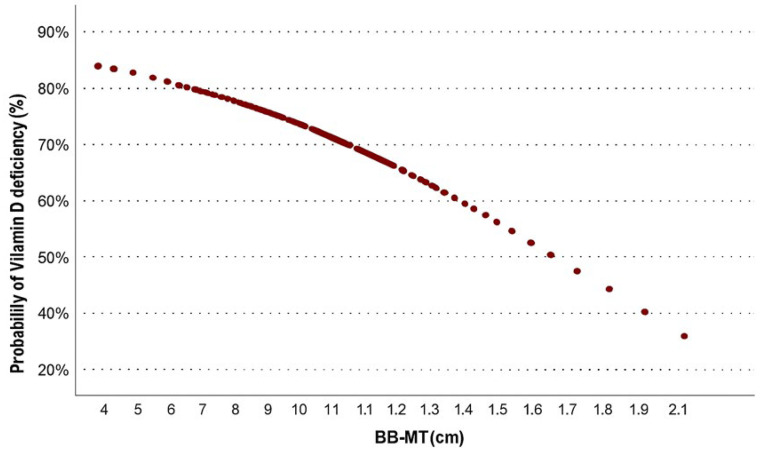
Probability of vitamin D deficiency by biceps brachii muscle thickness (BB-MT).

**Table 1 healthcare-13-02188-t001:** Descriptive characteristics of the patients.

Patients (n = 187)		Mean ± SD ^a^	Number (%)
		Median (Min-Max) ^b^	
Age ^a^		75.1 ± 14.4	
Sex (n, %)	female		104 (55.6%)
	male		83 (44.4%)
ADLs (n,%)	independent		13 (7%)
	dependent		174 (93%)
DM (n, %)	no		130 (69.5%)
	yes		57 (30.5%)
Cancer (n, %)	no		137 (73.3%)
	yes		50 (26.7%)
CVO (n, %)	no		129 (69%)
	yes		58 (31%)
Dementia (n, %)	no		121 (64.7%)
	yes		66 (35.3%)
HT (n, %)	no		88 (47.1%)
	yes		99 (52.9%)
BMI (kg/m^2^) ^b^		22.1 (15.6–44.4)	
CC (cm) ^b^		28 (13–45)	
MUAC (cm) ^b^		24 (13–46)	
25-0H vitamin D (nmol/L) ^b^		13 (3–58)	
Ultrasonography measurments			
RF-SFT (cm) ^b^		0.66 (0.1–2.6)	
RF-MT (cm) ^b^		0.56 (0.2–1.5)	
RF-MT/SFT ^b^		0.89 (0.2–3.0)	
RF-CSA (cm^2^) ^b^		1.57 (0.3–4.7)	
BB-SFT (cm) ^b^		0.3 (0.1–1.3)	
BB-MT (cm) ^b^		0.9 (0.4–2)	
BB-MT/SFT ^b^		3.0 (0.86–8.0)	
BB-CSA (cm^2^)^b^		2.6 (0.8–6.7)	

Abbreviations: ADLs = Activities of Daily Living; BB-CSA = biceps brachii cross-sectional area; BB-MT = biceps brachii muscle thickness; BB-MT/SFT = biceps brachii muscle thickness to subcutaneous fat thickness ratio; BB-SFT = biceps brachii subcutaneous fat thickness; BMI = body mass index; CC = calf circumference; CVO = cerebrovascular accidents; DM = diabetes mellitus; HT = hypertension; MUAC = mid-upper arm circumference; RF-CSA = rectus femoris cross-sectional area; RF-MT = rectus femoris muscle thickness; RF-MT/SFT = rectus femoris muscle thickness to subcutaneous fat thickness ratio; RF-SFT = rectus femoris subcutaneous fat thickness; SD = standard deviation; ^a^ = mean ± standard deviation; ^b^ = median (minimum–maximum).

**Table 2 healthcare-13-02188-t002:** Comparison of descriptive characteristics between the vitamin D groups.

		Vitamin D		
		Control Group (n = 60)	Deficient Group (n = 127)	*p*
Age ^a^		76 ± 13.7	74.7 ± 14.9	0.558
Sex (n, %)	female	40 (66.7%)	64 (50.4%)	0.037 ^χ2^
	male	20 (33.3%)	63 (49.6%)	
ADLs (n, %)	independent	10 (16.7%)	3 (2.4%)	<0.001 ^m^
	dependent	50 (83.3%)	124 (97.6%)	
DM (n, %)	no	38 (63.3%)	92 (72.4%)	0.207
	yes	22 (36.7%)	35 (27.6%)	
Cancer (n, %)	no	47 (78.3%)	90 (70.9%)	0.218
	yes	13 (21.7%)	37 (29.1%)	
CVO (n, %)	no	44 (73.3%)	85 (66.9%)	0.377
	yes	16 (26.7%)	42 (33.1%)	
Dementia (n, %)	no	39 (65%)	82 (64.6%)	0.954
	yes	21 (35%)	45 (35.4%)	
HT (n, %)	no	22 (36.7%)	66 (52%)	0.05 ^χ2^
	yes	38 (63.3%)	61 (48%)	
BMI (kg/m^2^) ^b^		23.5 (17–38.1)	22 (15.6–44.4)	0.02 ^m^
CC (cm) ^b^		29 (13–45)	27 (18–39)	0.006 ^m^
MUAC (cm) ^b^		26 (13–46)	24 (15–40)	0.057
25-OH vit D (nmol/L) ^b^		61.2 (50–145)	22.5 (7.5–47.5)	<0.001 ^m^
Ultrasonography measurements				
RF-SFT (cm) ^b^		0.84 (0.1–2.6)	0.59 (0.1–2)	<0.001 ^m^
RF-MT (cm) ^b^		0.65 (0.2–1.5)	0.53 (0.2–1.5)	<0.001 ^m^
RF-MT/SFT ^b^		0.810 (0.25–1.77)	0.920 (0.22–3.0)	0.049 ^m^
RF-CSA (cm^2^) ^b^		1.89 (0.3–4.5)	1.45 (0.4–4.7)	<0.001 ^m^
BB-SFT (cm) ^b^		0.33 (0.1–1.3)	0.3 (0.1–1)	0.210
BB-MT (cm) ^b^		0.65 (0.2–1.5)	0.5 (0.2–1.5)	0.009 ^m^
BB-MT/BB-SFT		3.07 (0.86–7.94)	3.06 (0.86–8.03)	0.968
BB-CSA (cm^2^) ^b^		3.05 (1.4–5.8)	2.6 (0.8–6.7)	0.002 ^m^

Abbreviations: ADLs = Activities of Daily Living; BB-CSA = biceps brachii cross-sectional area; BB-MT = biceps brachii muscle thickness; BB-MT/SFT = biceps brachii muscle thickness to subcutaneous fat thickness ratio; BB-SFT = biceps brachii subcutaneous fat thickness; BMI = body mass index; CC = calf circumference; CVO = cerebrovascular accidents; DM = diabetes mellitus; HT = hypertension; MUAC = mid-upper arm circumference; RF-CSA = rectus femoris cross-sectional area; RF-MT = rectus femoris muscle thickness; RF-SFT = rectus femoris subcutaneous fat thickness; RF-MT/SFT = rectus femoris muscle thickness to subcutaneous fat thickness ratio; ^a^ = mean ± standard deviation; ^b^ = median (minimum–maximum); ^m^ = the Mann–Whitney U test; ^χ2^ = the Chi-square test.

**Table 3 healthcare-13-02188-t003:** Results of logistic regression analysis of vitamin D groups.

		Univariable						Multivariable				
	β	OR	%95 CI			*p*	β	OR	%95 CI			*p*
Sex	0.677	1.96	1.04	-	3.73	0.038 *	-	-	-		-	
Age	−0.006	0.99	0.972	-	1.02	0.57	-	-	-		-	
ADL dependency	2.122	8.26	2.183	-	31.3	0.002 *	2.182	8.86	2.28	-	34.43	0.002 *
HT	−0.625	0.53	0.285	-	1.01	0.052	-	-	-		-	
RF-MT/SFT	0.920	2.51	1.179	-	5.344	0.017 *	0.994	2.70	1.21	-	6.03	0.015 *
BB-MT	−1.307	0.27	0.094	-	0.779	0.015 *	-	-	-		-	

Abbreviations: ADL = Activities of Daily Living; BB-MT = biceps brachii muscle thickness; HT = hypertension; RF-MT/SFT = rectus femoris muscle thickness to subcutaneous fat thickness ratio; OR = odds ratio; CI = confidence interval; * = *p* < 0.05. Model variables: dependent variable = vitamin D deficiency; independent variables = gender, age, ADL dependency, HT, RF-MT/SFT, BB-MT. The Hosmer–Lemeshow test: *p* = 0.931. Forward LR = forward logistic regression.

## Data Availability

The data presented in this study are available from the corresponding author upon reasonable request.

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
