# Peer review of "Association of Vitamin D Deficiency with Local Muscle–Fat Ratio in Geriatric Palliative Care Patients: An Ultrasonographic Study"

_healthcare, 2025, doi:10.3390/healthcare13172188_

Round 1
Reviewer 1 Report
Comments and Suggestions for Authors
Review of paper “Association of Vitamin D Deficiency with Muscle–Fat Ratio in Geriatric Palliative Care Patients: An Ultrasonographic Study”. This is a cross-sectional study, investigating vitamin D status and Ultrasound measures of muscle mass in older adults. I believe that this paper should improve its introduction and conclusion reporting and pay more attention to statistical analysis to ensure its publication.
- Authors should prefer to use “age-inclusive” language, and maybe avoid using “elderly person” in their text.
- abstract, conclusion: It is hard to agree with a conclusion that suggests “supplementation as a strategy to support physical function” when the data is cross-sectional and prone to reverse causation. I would recommend that the authors conclude that prospective cohort studies and supplementation studies are necessary.
- abstract, line 16: is it <20 or <50 nmol/L?
- Introduction, lines 67-69: “Vitamin D is a modifiable and easily accessible biomarker that is often underrecognized in clinical practice, yet it may have meaningful implications for maintaining physical function in this high-risk population”. I do not quite agree with such a statement. Vitamin D has been extensively studied in several aspects of clinical practice, including older adults. Vitamin D supplementation and vitamin D blood levels test have been widely (and potentially wrongly) recommended worldwide for basically every individual. Authors should rethink such a statement in their introduction, in my opinion.
- Introduction: Authors should consider incorporating evidence from cohort studies demonstrating a prospective association between vitamin D levels and muscle strength. and try to put their paper in a better perspective, improving their introduction section, showing the gaps in this research area
- sample size calculation: the effect size of 0.37 was for which association, specifically. Authors must report the exact association for which the sample size was calculated. Was it from a t-test?
- sample size: it is awkward to consider 3% losses; usually, researchers consider 10% of losses. Is there any reference backing up this choice?
- line 151, statistical analysis: authors actually used “Multivariable models” and not “multivariate models”. Please, adjust it throughout the text.
- The use of the “forward” method for selecting the most appropriate models is not advisable. Please see Smith, G. Step away from stepwise. J Big Data 5, 32 (2018). https://doi.org/10.1186/s40537-018-0143-6. Ideally, authors should use other methods for defining the best model, such as comparing the AIC of the models with each term.
- I question the choice of logistic models. In the logistic model, vitamin D status is the OUTCOME. However, it makes no physiological sense to suggest that low muscle mass leads to vitamin D deficiency; it is probably the other way around, so linear models should have been used by the authors.
- figure 3. The y-axis is not in English language.
- Conclusion: I suggest authors refrain from concluding cause and effect when none can be inferred from this study.
Author Response
The responses provided by the reviewers have been uploaded as a file.

Reviewer 2 Report
Comments and Suggestions for Authors
Please check the attached file.

Author Response
Please see the attached file for detailed responses to reviewers."

Round 2
Reviewer 1 Report
Comments and Suggestions for Authors
The authors satisfactorily answered all my comments. Although I still disagree with the statistical approach chosen by the authors (stepwise regression, and vitD status as outcome in a logistic model), the authors acknowledged such choices in the limitations section of their paper.
Author Response
The reviewers' comments have been included as a separate Word file.

Reviewer 2 Report
Comments and Suggestions for Authors
Dear Authors,
I have reviewed your responses to the first round of review and confirm that the issues and questions raised during the initial evaluation have been satisfactorily addressed.
Thank you for your diligent work on the revisions.
However, I recommend that you carefully proofread the entire manuscript to correct any typographical errors. Addressing these will further enhance the overll quality of your manuscript.
Author Response
The reviewers' comments have been included as a separate PDF file
